# The Lipid-Soluble Bioactive Substances of *Fagopyrum esculentum* Varieties under Different Tillage and Nitrogen Fertilisation

**DOI:** 10.3390/foods11233801

**Published:** 2022-11-25

**Authors:** Krzysztof Dziedzic, Szymon Kurek, Grażyna Podolska, Sławomira Drzymała-Czyż, Sylwia Mildner-Szkudlarz, Wei Sun, Jarosław Walkowiak

**Affiliations:** 1Department of Food Technology of Plant Origin, Poznań University of Life Sciences, Wojska Polskiego 28, 60-637 Poznań, Poland; 2Department of Pediatric Gastroenterology and Metabolic Diseases, Institute of Pediatrics, Poznan University of Medical Sciences, Szpitalna 27/33, 60-572 Poznań, Poland; 3Institute of Soil Science and Plant Cultivation State Research Institute, Czartoryskich 8, 24-100 Puławy, Poland; 4Department of Bromatology, Faculty of Pharmacy, Poznan University of Medical Sciences, Rokietnicka 3, 60-806 Poznań, Poland; 5Key Laboratory of Beijing for Identification and Safety Evaluation of Chinese Medicine, Institute of Chinese Materia Medica, China Academy of Chinese Medical Sciences, Beijing 100700, China

**Keywords:** common buckwheat, phytosterols, squalene, tocopherols, plow tillage, cultivars

## Abstract

Yield and grain composition play an important role in food production. The aim of this study was to determine the content of phytosterols, squalene, and tocopherols in four varieties of common buckwheat grains: Kora, Panda, Smuga, and Korona depending on the tillage and nitrogen doses employed. The grains were cultivated at the Experimental Station of the State Research Institute of Soil Science and Plant Cultivation in Osiny, Poland, with plow tillage, without plow tillage cultivation, and with nitrogen fertilizers (0, 50, and 100 kg N_2_ ha^−1^). Gas chromatography with a mass spectrometer was used to assess all the parameters studied. The cultivation methods did not affect the levels of phytosterols, tocopherols, and squalene, but the highest levels of phytosterols were seen in the Kora variety with 50 kg N_2_ ha^−1^ (ranging from 1198 μg to 1800 μg·g^−1^ of sample weight); therefore, the variety was the best source of phytosterols investigated.

## 1. Introduction

Recent years have seen an increase in the number of conscious consumers, who pay particular attention to the quality of food products, expecting them to be free of herbicides and pesticides and also to have good nutrient values. Buckwheat is a pseudocereal that does not require the extensive use of chemical substances for its cultivation, and so can easily be considered an ecological plant. Buckwheat is used in gluten-free products as a substitute for gluten flour and can be used together with corn or rice flour to increase the nutritional value of gluten-free bread [1]. Common buckwheat (*Fagopyrum esculentum*) contains dietetic starch (70–80%), dietary fiber (10–12%), water-soluble B-group vitamins, flavonoids (mainly rutin and quercetin), proteins with well-balanced amino acids (10–12%) and lipids (2%) of high nutritional value [2,3,4]. The amylose content of common buckwheat starch granules is higher than in other cereals, so buckwheat flour has the potential for producing food with a low glycemic index [5]. The proteins found in common buckwheat have low digestibility, on account of the high levels of flavonoids, which block gastric enzymes; however, this does not decrease the biological value of the proteins [5]. Furthermore, the proteins reduce blood cholesterol levels. Lipids and lipid-soluble substances have great biological value in human nutrition. High concentrations of squalene, phytosterols, and tocopherols can positively affect human blood after long and systematic consumption. For all these reasons, common buckwheat is worth considering as a source of functional foods aimed at aware consumers, foods that could be used to prevent diseases of affluence [3,5].

Nutrient levels in plant food can be increased using various varieties of plants, different cultivation methods, various fertilization schemes, and different technological processing regimes [6]. In the last decade, innovations in the breeding and cultivation of cereals and pseudocereals have yielded increased nutrients. Different levels of organic fertilizers or chemicals such as phosphate salts, nitrogen salts, magnesium salts, iron salts, and zinc have been used during cultivation [7,8,9,10]. There are several tillage systems: plow tillage, no-till tillage, and no tillage. The trend of moving away from plowing, which is energy-consuming and causes soil degradation, is increasingly observed. The effects of different methods on the wheat quality show varying results; however, many authors have described their important role regarding protein content, hectolitre mass, gluten, and particle size index of wheat, therefore it seems to be justified to test other crop cultivation [11]. Such innovations aim to meet consumer expectations regarding levels of prohealth substances in the final food product. Grain yield benefits are also important for farmers, especially given the changing climate. Optimization of the commercial cultivation of buckwheat in Europe is therefore necessary, using a variety of approaches to breeding and cultivation. According to FAOSTAT (2020), the annual world production of buckwheat is 1.81 million tons but could be increased [12].

One of the most important agrotechnical factors affecting the yield of grain is fertilization by minerals, especially nitrogen salts. In intensive plant production, nitrogen (N) fertilization is essential for plant productivity and good-quality yield [13]. The number of studies on nitrogen fertilization and the cultivation requirements of buckwheat remains limited. The recommended nitrogen level per hectare for buckwheat is 50 kg, and an additive correlation between grain production and protein level has been seen [14]. However, there is still a lack of evidence for the effects of nitrogen fertilization on the profile of some of the bioactive lipids-soluble substances in buckwheat. Taking into consideration the information mentioned above about phytosterols, squalene, and tocopherols, it is reasonable to study the effects of cultivation (with and without plow tillage) and fertilization (without nitrogen, and with nitrogen at 50 kg·ha^−1^ and 100 kg·ha^−1^) on the levels of bioactive substances in four registered varieties of common buckwheat: Kora, Panda, Smuga, and Korona (PA-15).

## 2. Material and Methods

### 2.1. Reagents and Standards

The chemicals were obtained from Poch (Gliwice, Poland). These were potassium hydroxide (1 M methanolic solutions), hexane, Methyl tert-butyl ether (MTBE), methanol, potassium hydrogen phosphate, and petroleum ether. Standards of tocopherol (α-, β-, δ- and γ-tocopherol), pyridine and 5-alpha-cholestan, and Tween 20 were obtained from Sigma Aldrich (Darmstadt, Germany). BSTFA + TMCS, 99:1 was purchased from Supelco (Munich, Germany).

### 2.2. Plant Material and Experimental Design

A field experiment was established in season 2019 at the Experimental Station of the State Research Institute of Soil Science and Plant Cultivation (IUNG-PIB) in Osiny, Poland (51°28′45 N 22°03′16 E). The buckwheat (*Fagopyrum esculentum*) was sown on the pseudopodsolic soil typical of the region (winter wheat was the forecrop), with extractable phosphorus (P: 9.54 mg kg^−1^), exchangeable potassium (K: 12.0 mg kg^−1^), and pH KCl 6.4. Buckwheat was sown at a seed rate of 2.5 million ha^−1^ on 6 May 2019. Mineral fertilizers (kg/ha, P_60_K_90_) were applied throughout the experimental field before sowing. The harvest took place on 16 September 2019. No pesticides were applied during the cultivation. The field experiment was designed with a split-plot approach in three replications (plot size 6.0 × 6.0 m). The first factor was soil tillage (plow tillage or lack thereof), conventional tillage applied in autumn: cultivation with a disc cultivator (2×), plowing with a 4-furrow plow. In spring harrowing with heavy harrow cultivation with a disc cultivator pre-sowing cultivation with an aggregate. No-plow tillage applied in autumn: cultivation with a disc cultivator (3×). In spring: harrowing with a heavy harrow, cultivation with a disc cultivator, pre-sowing cultivation with an aggregate. The second was nitrogen fertilization dose (0, 50, 100 kg/ha N_2,_ ammonium sulfate—50%/50%, NO_3_^−^/NH_4_^+^), and the third was cultivar (Kora, Panda, SMH Smuga, and SMH Korona PA15). All samples were chipped for analysis using a laboratory mill (Sadkiewicz, type MH-10a, Bydgoszcz, Kuyavian-Pomeranian, Poland <380 μm). The samples were examined in triplicate.

### 2.3. Sterols, Tocopherols, Squalene, and Cholesterol Contents

Phytosterols, squalene, and tocopherol levels were analyzed following the AOCS Official Method [15,16]. Directly in the sample (about 0.5 g) 5-alpha-cholestan (130 μL, concentration 1 mg·mL^−1^) was added as an internal standard. The extraction of oil was conducted with Folch solvent (7 mL of methanol/chloroform 1:1). Horizontal shaker (mrc, Essex, East of England, UK), was used for this process and the time of extraction was 2 h. The samples were then dried under a stream of nitrogen. The amount of extracted oil was not determined. All of the dry residue after solvent evaporation was used for further analysis. The saponification process was performed over 18 h with potassium hydroxide solution. In the next step, the hexane/MTBE mixture was used to extract the non-saponifiable fraction, and the samples were dried using a stream of nitrogen. The BSTFA with 1% TMCS was placed and the samples were incubated at 80 °C to enhance the silylation reaction. The prepared samples were moved to microvials and analyzed by GC (gas chromatography). An Agilent 7890A gas chromatograph (Agilent, San Jose, California, USA) coupled with a 5975C mass selective detector (MSD) was used (HP-5ms capillary column, 30 m × 0.25 mm × 0.25 μm). The analysis parameters were as follows: the temperature of analysis was programmed (starting at 250 °C held for 2 min, then rising to 265 °C at 2 °C·min^−1^, where it was held for 9 min, and finally rising to 275 °C at 1 °C·min^−1^, where it was held for 10 min), and the temperatures of the injection port and FID detector were set to 290 °C and 300 °C, respectively. Helium with a flow rate of 1.15 mL·min^−1^ was used. The sample was injected with a split ratio of 20:1. We used a flame ionization detector(Agilent, San Jose, California, USA) for quantitative determination, and the MSD confirmed the identity of the sterols (ISO 12228-1:2014). The oven temperature program was the same as for the FID analysis. The source and quadrupole temperatures were 230 °C and 150 °C, respectively. The analysis was performed in SCAN mode (50–550 u). The tocopherols and squalene were identified on the basis of external standards. Sterols and cholesterol were initially identified by chromatogram comparison of the current samples with previously analyzed samples and other well-described oils in the literature (sunflower oil, olive oil, etc.). Quantitation was carried out using internal standards (Figure 1).

### 2.4. Statistical Analysis

All samples were analyzed in triplicate. One-way analysis of variance (Anova) was used. A post hoc (Tukey’s) test to determine the significance of differences between means was conducted. Dependencies were considered statistically significant at α < 0.05. Principal component analysis (PCA) was performed to reduce the dimensionality of data and to present the samples in the coordinate system. All calculations were carried out with Statistica software (Statistica 13.3, StatSoft, Tulsa, OK, USA).

## 3. Results

### 3.1. Buckwheat Grain Yield

In our research, the grain yield of buckwheat was significantly (*p* ≤ 0.05) affected by nitrogen fertilization level and cultivar. Nitrogen fertilization at 100 kg ha^−1^ increased grain yield by ~0.117 t ha^−1^ on average compared to the 50 kg·ha^−1^ N_2_ level, and by ~0.311 t ha^−1^ compared to no nitrogen fertilization. The cultivar with the greatest yield was MHR Smuga (1.321 t ha^−1^), MHR Korona PA-15 (1.265 t ha^−1^), Panda (1.262 t ha^−1^), and MHR Kora (1.23 t ha^−1^). There were no significant differences in yield level between the plow and no-plow tillage.

### 3.2. The Effect of Varieties, Plow Tillage, and Nitrogen on Phytosterol Content

The absence or presence of plow tillage in the used varieties and doses of nitrogen affected the total phytosterol content in buckwheat grains (Table 1).

The highest concentration of total phytosterols was found in the Kora samples (from 1138 μg to 1800 μg·g^−1^ of samples), and the lowest in the Smuga variety (from 917 μg to 1183 μg·g^−1^ of samples). We identified seven phytosterols in all the varieties of common buckwheat: campesterol, stigmasterol, β-sitosterol, d5-avenasterol, 5,24-stigmastadienol, 7-stigmastenol, and d7-avenasterol (Table 2).

The highest concentration of β-sitosterol was seen in Kora varieties without plow tillage and with 50 kg·ha^−1^ nitrogen (1302 μg·g^−1^ of samples), and the lowest level was seen in the Smuga variety cultivated with 50 kg·ha^−1^ nitrogen (746 μg·g^−1^ of samples). The highest level of d5-avenasterol and campesterol in the samples was found in the Kora variety without plow tillage and with 50 kg·ha^−1^ nitrogen (161 and 147 μg·g^−1^ of samples, respectively), while the lowest level was seen for the no-plow tillage Smuga sample with 100 kg·ha^−1^ of nitrogen (82 and 67 μg·g^−1^ of samples, respectively). The highest amounts of stigmasterol and stigmastadienol were found in untilled Kora with 50 kg nitrogen (43 and 27 μg/g of samples, respectively); however, the lowest concentrations were observed in untilled Smuga without nitrogen fertilizer (21 and 11 μg/g of samples, respectively). We did not observe a similar tendency for 7-stigmastenol and d7-avenasterol. The highest concentration of 7-stigmasterol was seen in no-plow tillage Kora varieties with 50 kg of nitrogen (60 μg·g^−1^ of samples), and the lower level in no-plow tillage Smuga with 100 kg of nitrogen (32 μg·g^−1^ of samples). The plow tillage Kora without nitrogen supplements contained the highest amounts of d7-avenasterol (13 μg·g^−1^ of samples), while the PA-15 varieties without plow tillage and without nitrogen supplementation also had the lowest concentration of d7-avenasterol (8 μg·g^−1^ of samples).

### 3.3. The Effect of Varieties, Plow Tillage, and Nitrogen on Tocopherol Content

Plow tillage or its absence did not lead to any differences in tocopherol concentration (Table 3).

The total tocopherol content in the Smuga and Kora varieties was higher (30–47 μg·g^−1^ of samples in both varieties) than in the Panda variety (20–35 μg·g^−1^ of samples; Table 1). γ-Tocopherol dominated in plow tillage Kora with 50 kg·ha^−1^ of N_2_ (36 μg·g^−1^ of samples), no-plow tillage Kora with 50 and 100 kg of N_2_ (41 and 37 μg·g^−1^ of samples, respectively), and no-plow tillage Smuga with 50 kg·ha^−1^ of N_2_ (41 μg·g-1 of samples). The highest concentration of δ-tocopherol in the samples was found in no-plow tillage Kora with 50 kg·ha^−1^ N_2_ (4.9 μg·g^−1^ of samples), Smuga with 50 kg ha^−1^ N_2_, and Pa-15 with 100 kg·ha^−1^ N_2_ (3 μg·g^−1^ of samples in both cases). α-Tocopherol levels were highest in the no-plow tillage Smuga variety (50 and 100 kg·ha^−1^ of N_2_), the Pa-15 variety (100 kg·ha^−1^ of N_2_), the no-plow tillage Kora (50 N_2_ kg·ha^−1^), the Panda variety (0 N_2_ kg·ha^−1^), the Smuga variety (50 and 100 N_2_ kg·ha^−1^), and the Pa-15 variety (0 and 50 N_2_ kg ha^−1^), and fluctuated between 0.6 and 0.8 μg·g^−1^ of samples. The highest level of β-tocopherol was seen in the no-plow tillage Kora variety with 50 kg·ha^−1^ N_2_ and in the plow tillage Kora without N_2_ (0.31 and 0.28 μg·g^−1^ of samples, respectively). We observed the differences in the four varieties of buckwheat regarding α-tocopherol, γ-tocopherol, and total tocopherol concentration. α-Tocopherol in the Panda variety (the smallest amounts) differed in comparison to Kora, Smuga, and PA-15. Thus, we can recommend not using the Panda variety if a high concentration of α-tocopherol is desired (Table 3). The highest levels of γ-tocopherol and total tocopherols were seen in the Kora and Smuga varieties.

### 3.4. The Effect of Varieties, Plow Tillage, and Nitrogen on Squalene and Cholesterol Content

Plow tillage or its absence did not lead to differences in the squalene or cholesterol content in the case of the Panda, Smuga, and Pa-15 varieties (Table 1). No association was determined between varieties, cultivation methods, and nitrogen fertilization regarding squalene; its concentration ranged from 10.4 μg·g^−1^ of samples (PA-15, without N_2_, with tillage) to 57.7 μg·g^−1^ of samples (Kora, 80 kg·ha^−1^ N_2_, no tillage). We only observed an effect of plow tillage on squalene content when the Kora variety was fertilized by three doses of nitrogen (Table 1). No effect on cholesterol level was seen for variety, fertilization, or cultivation method.

## 4. Discussion

### 4.1. Buckwheat Grain Yield

We observed an increase in grain yield after nitrogen fertilization. A similar tendency has been observed by other authors when nitrogen fertilization was increased from 0 to 30 kg·ha^−1^ N_2_, giving a final grain yield of about 1.54 t ha^−1^ [13]; however, the authors with a higher application of nitrogen observed grain scattering and lodging. Presented research on the grain yield of two cultivars of common buckwheat (Hruszowska and Prego) also showed differences, and the more significant effect of nitrogen fertilizer was observed in the case of the Hruszowska variety [14].

### 4.2. The Effect of Varieties, Plow Tillage, and Nitrogen on Phytosterol Content

Plant sterols are good cholesterol-lowering agents, with consumption of 1.5 g to 3 g phytosterols per day reducing the blood plasma concentrations of low-density cholesterol lipoprotein (LDL-c) by up to 12% [17]. The consumption of 2 g·day^−1^ phytosterols with food can contribute lower the risk of diseases of affluence, such as coronary heart disease and cancer [18]; the consumption of buckwheat products is thus highly recommended. The literature has limited information on the levels of phytosterols in varieties of buckwheat. The effects of thermal processing of the grains (Kora variety) have previously been described. One group of researchers documented that raw buckwheat grains have the lowest total phytosterol content (530 μg·g^−1^ of samples), but this may be related to weather conditions, as noted in 2014 [2]. In another experiment described in 2016, raw Kora buckwheat grains were shown to contain 690 μg·g^−1^ of samples of the phytosterols [19]. This indicates that many factors, including weather, soil, fertilization, and cultivation, can affect the total phytosterol content. It has previously been suggested that β-sitosterol is also the dominant sterol in the Kora variety [2,19] and, as the most abundant sterol, accounts for 50–70% of total sterols [20]. Raguindin et al. described that campesterol came immediately after β-sitosterol in abundance in buckwheat and oats [20]. Dziedzic et al. also showed that the order of the phytosterol concentrations in the Kora variety from dominant to least was sitosterol > campesterol > d7-stigmasterol > cycloartenol > stigmasterol > avenasterol > sitostanol [19]. Previous research identified stigmasterol and avenasterol in Kora variety grains, but stigmastadienol or 7-stigmastenol were not identified [19]. Phytosterols are present in all parts of buckwheat, both green parts and in all layers of the grains. Its concentration significantly varies with the tissue and growing phase [4]. The previous study did not describe how the concentration of phytosterols in different genotypes of common buckwheat depends on the methods of cultivation and fertilization. Alignan et al. observed that there was a significant effect of bread wheat genotype and weather conditions on phytosterol and phytostanol content. This result showed that the genotypic factor dominated the sowing date factor in determining sterol and stanol traits in bread wheat cultivated under organic conditions. [21]. Chung et al. documented that the concentration of β-phytosterol in sorghum grains differed by genotype and cultivation areas [22]. Other authors have shown that drought stress does not affect the levels of phytosterols in different genotypes of *Foeniculum vulgare* grain [23]. We can thus conclude that phytosterol diversity may depend on many of the factors mentioned here. However, for optimizing the total phytosterol content we recommend no-plow tillage Kora variety with 50 kg·ha^−1^ N_2_ of fertilization.

### 4.3. The Effect of Varieties, Plow Tillage, and Nitrogen on Tocopherol Content

A previous study examined α-, β-, γ-, and δ-tocopherol composition of sorghum grains with different genotypes and cultivation areas, observing differences in the total levels of phytosterols depending on the variety and the growth location (Chung et al., 2013). No statistical differences were observed in the tocopherol composition for N_2_ fertilizers of walnuts. Differences were seen only for α-tocopherol, and the observed tendency of dependency on investigated variety was similar to that observed in our study [24].

### 4.4. The Effect of Varieties, Plow Tillage, and Nitrogen on Squalene and Cholesterol Content

No significant effects of plow tillage on the protein, oil, and mineral content of corn grain were observed [25]. Other authors have indicated that the number of starch granules in nitrogen-cultivated Tartary buckwheat grains was higher than under normal cultivation conditions, but there remains a lack of evidence for the effects of nitrogen fertilization and plow tillage on the lipid composition of buckwheat grains. On the other hand, organic and conventional cultivation did not lead to a difference in common buckwheat secondary metabolites such as flavonoids [26,27]. Compared with no nitrogen, the optimum level of nitrogen fertilizer reduced the total lipids of oat grains by a small but nonetheless significant amount [28]. Kalinova et al. described the squalene distribution and concentration of three varieties of buckwheat leaves and flowers in different stages of plant growth, finding the highest level of squalene in the leaves (from 84 to 98 μg·g^−1^ of samples, depending on variety). They also noted that squalene and other secondary metabolites, such as tocopherols, are distributed from the leaves to grains [27]. Insignificant levels of cholesterol derivatives have been identified in common buckwheat, but their nutritional importance is minor [20].

### 4.5. Principal Component Analysis (Plow Tillage, Fertiliser, Varieties)

The results of the PCA analysis of the varieties and other variables are shown in Figure 2.

When all the variables (phytosterol levels, tocopherol levels, squalene levels, cholesterol levels, fertilization doses, and plow tillage) are considered, two outliers, relating to the Kora (red circle) and Smuga varieties (green circle), can be distinguished. There were no differences between samples related to cultivation or fertilization methods. Verardo et al. also studied the effects of N_2_ fertilization on the phytosterol content of walnuts and showed that N_2_ fertilization did not affect phytosterol levels in the nuts [29]. PC2 correlated positively mainly with phytosterols and with γ- and δ-tocopherols, while the correlation was negative for squalene, α- and β-tocopherol levels, as well as fertilization doses. Phytosterols’ prohealth activity seems to be a good feature of buckwheat, which is a valuable source of phytosterols, as cultivation and N_2_ fertilization have a small effect on phytosterol content regarding four examined varieties.

## 5. Conclusions

Nitrogen fertilization differently affects the level of phytosterols, tocopherols, squalene, and cholesterol, depending on the buckwheat variety; however, the cultivation method (plow tillage) has the smallest effects on lipid quality. Among the analyzed grains of four buckwheat varieties, the best source of phytosterols is the Kora variety, while the seeds of the Smuga variety are the least recommended. Kora and Smuga grains contain the highest content of tocopherols. The phytosterol content in the Kora variety with 50 kg ha^−1^ of nitrogen was higher by 33% in comparison to crops without nitrogen. In Smuga varieties an increase of 20% of total phytosterols between 100 and 50 kg ha^−1^ nitrogen fertilizer was observed. We recommend the cultivation of Kora varieties in order to give better lipid composition. For farmers, from an economical point of view, and taking into consideration lipid-soluble substances and buckwheat grain yield, the no-plow tillage cultivation of Kora varieties with 50 kg/ha of nitrogen fertilization is highly recommended. Grains of buckwheat in the Kora variety can be used as a source of bioactive lipid-soluble substances for functional food production.

## Figures and Tables

**Figure 1 foods-11-03801-f001:**
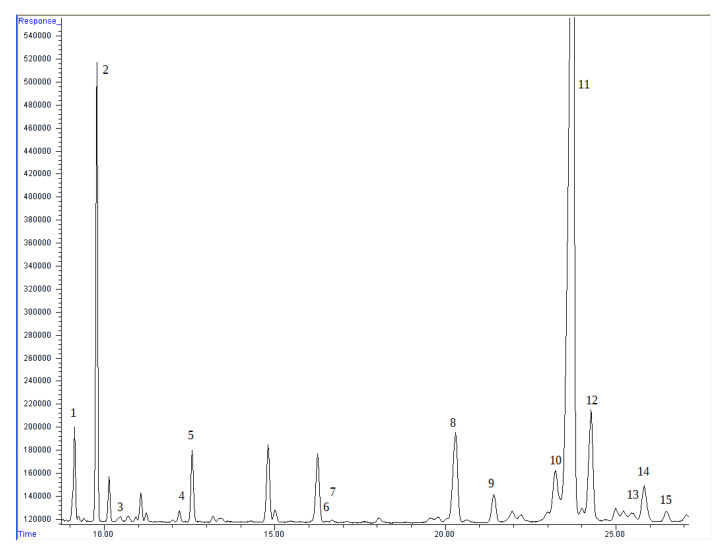
Chromatogram of detected lipid-soluble substances. 1: squalene; 2: 5-alpha-cholestan (internal standard); 3: delta-tocopherol; 4: beta-tocopherol; 5: gamma-tocopherol; 6: alpha-tocopherol; 7: cholesterol; 8: campesterol; 9: stigmasterol; 10: clerosterol; 11: beta-sitosterol; 12: d5-avenasterol; 13: 5,24-stigmastadienol; 14: 7-stigmastenol; 15: d7-avenasterol.

**Figure 2 foods-11-03801-f002:**
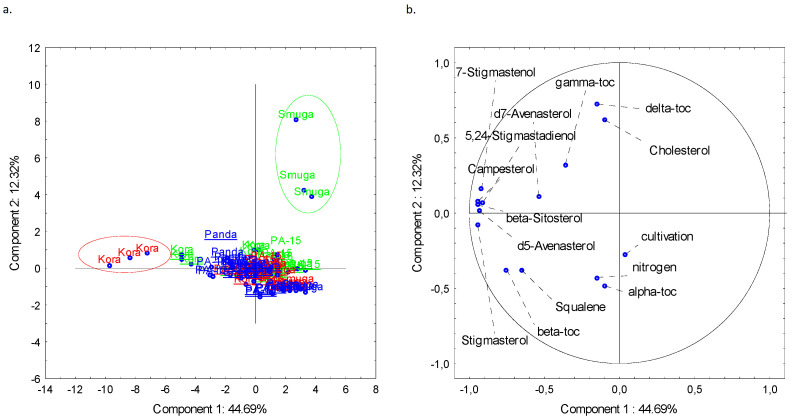
Two-dimensional plot representing the PCA: loading plot (**a**), and score plot (**b**). Green color-fertilization with 0 kg/ha of N_2_; red color-fertilization with 50 kg/ha of N_2_; blue color-fertilization with 100 kg/ha of N_2_; Underline-cultivation with plow tillage, toc-tocopherol.

**Table 1 foods-11-03801-t001:** Total phytosterol, tocopherol, squalene, and cholesterol content of four varieties of no-plow tillage and plow tillage common buckwheat, by fertilization dose.

Varieties	Dose of N (kg/ha)	Total Phytosterol Content (μg/g)	Total Tocopherol Content (μg/g)	Squalene (μg/g)	Cholesterol (μg/g)
No-plow tillage					
Kora	0	1213 ^defg^ ± 7	37.37 ^de^ ± 0.11	18.50 ^jk^ ± 0.82	5.01 ^d^ ± 0.02
	50	1800 ^a^ ± 78	47.12 ^ab^ ± 1.12	57.65 ^a^ ± 1.68	3.11 ^e^ ± 0.11
	100	1307 ^de^ ± 57	40.64 ^bcd^ ± 0.49	36.68 ^defg^ ± 1.20	1.73 ^ghi^ ± 0.25
Panda	0	1176 ^efghi^ ± 71	20.12 ^j^ ± 0.29	37.55 ^cde^ ± 1.68	2.08 ^fghi^ ± 0.19
	50	1193 ^defgh^ ± 75	34.26 ^def^ ± 0.89	32.83 ^efgh^ ± 1.02	2.71 ^ef^ ± 0.30
	100	1103 ^ghi^ ± 42	25.91 ^hij^ ± 0.65	46.38 ^b^ ± 0.75	1.60 ^hi^ ± 0.07
Smuga	0	1052 ^hij^ ± 34	47.74 ^a^ ± 9.58	8.42 ^m^ ± 3.66	10.03 ^a^ ± 0.99
	50	1102 ^ghi^ ± 66	44.59 ^abc^ ± 1.25	30.91 ^fgh^ ± 1.56	1.60 ^hi^ ± 0.01
	100	917 ^j^ ± 30	37.82 ^de^ ± 0.42	30.49 ^ghi^ ± 0.83	1.29 ^i^ ± 0.05
PA-15 (Korona)	0	1119 ^ghi^ ± 32	35.41 ^def^ ± 0.93	15.06 ^kl^ ± 6.20	2.43 ^efgh^ ± 0.24
	50	1208 ^defg^ ± 41	36.64 ^def^ ± 0.53	34.80 ^efg^ ± 0.45	3.19 ^e^ ± 0.26
	100	1462 ^c^ ± 9	51.10 ^a^ ± 0.58	44.95 ^b^ ± 0.77	2.65 ^ef^ ± 0.32
Plow tillage					
Kora	0	1627 ^b^ ± 56	34.12 ^def^ ± 1.04	41.39 ^bcd^ ± 0.98	5.64 ^d^ ± 0.47
	50	1198 ^defgh^ ± 53	36.80 ^de^ ± 0.79	24.46 ^ij^ ± 1.75	2.69 ^ef^ ± 0.01
	100	1138 ^efghi^ ± 53	38.53 ^cde^ ± 0.48	43.42 ^bc^ ± 1.32	8.66 ^b^ ± 0.46
Panda	0	1192 ^defgh^ ± 78	34.62 ^def^ ± 0.2	33.47 ^efgh^ ± 0.49	2.61 ^def^ ± 0.15
	50	1287 ^def^ ± 49	35.13 ^def^ ± 1.04	27.65 ^hi^ ± 2.46	1.58 ^hi^ ± 0.06
	100	1341 ^cd^ ± 5	33.89 ^ef^ ± 0.18	30.68 ^fgh^ ± 1.92	6.89 ^c^ ± 0.09
Smuga	0	1118 ^ghi^ ± 60	30.31 ^fgh^ ± 1.13	20.37 ^jk^ ± 0.48	2.45 ^efgh^ ± 0.10
	50	1033 ^ij^ ± 15	37.91 ^de^ ± 0.33	28.64 ^hi^ ± 0.65	1.64 ^hi^ ± 0.15
	100	1183 ^efghi^ ± 30	34.09 ^ef^ ± 0.49	45.95 ^b^ ± 0.65	7.73 ^c^ ± 0.23
PA-15 (Korona)	0	1093 ^ghi^ ± 21	26.97 ^ghi^ ± 0.36	10.36 ^lm^ ± 1.44	2.14 ^fghi^ ± 0.09
	50	1150 ^efghi^ ± 58	32.58 ^efg^ ± 0.58	28.17 ^hi^ ± 1.66	1.98 ^fghi^ ± 0.16
	100	1211 ^defg^ ± 20	22.05 ^ij^ ± 0.68	35.45 ^defg^ ± 1.36	2.20 ^fgh^ ± 0.01

Data are mean values of triplicate determinations ± standard deviation; means with different letters in each column differ significantly (*p* < 0.05).

**Table 2 foods-11-03801-t002:** Phytosterol content of four varieties of no-plow tillage and plow tillage common buckwheat, by fertilization doses.

Varieties	Dose of N (kg/ha)	β-Sitosterol (μg/g)	d5-Avenasterol (μg/g)	Campesterol (μg/g)	Stigmasterol (μg/g)	5,24-Stigmastadienol (μg/g)	7-Stigmastenol (μg/g)	d7-Avenasterol (μg/g)
No-plow tillage								
Kora	0	869 ^cdefgh^ ± 8	112 ^fgh^ ± 1	100.6 ^cde^ ± 0.7	27.9 ^cdefg^ ± 0.3	15.5 ^c^ ± 0.2	40.9 ^defg^ ± 0.5	9.6 ^defgh^ ± 0.7
	50	1302 ^a^ ± 59	161 ^a^ ± 7	147.3 ^a^ ± 4.5	43.3 ^a^ ± 3.4	26.6 ^a^ ± 1.9	60.1 ^a^ ± 2.2	12.8 ^abc^ ± 0.5
	100	943 ^cd^ ± 44	121 ^def^ ± 5	103.1 ^cd^ ± 3.6	29.1 ^cdef^ ± 1.2	15.4 ^c^ ± 0.6	43.0 ^cdef^ ± 1.7	11.7 ^abcdef^ ± 0.4
Panda	0	872 ^cdefg^ ± 56	105 ^ghijk^ ± 7	89.6 ^fghij^ ± 5.0	26.1 ^efgh^ ± 1.8	14.4 ^cdef^ ± 0.6	39.2 ^fgh^ ± 2.0	9.2 ^fgh^ ± 1.3
	50	863 ^cdefgh^ ± 56	114 ^efg^ ± 8	91.1 ^efghi^ ± 4.7	28.5 ^cdef^ ± 1.7	13.9 ^cdef^ ± 1.1	38.7 ^fgh^ ± 2.7	10.3 ^cdefgh^ ± 0.5
	100	812 ^efghi^ ± 34	103 ^ghijkl^ ± 4	81.0 ^ijk^ ± 2.6	25.7 ^fgh^ ± 0.8	11.7 ^efgh^ ± 0.2	33.7 ^hi^ ± 0.3	9.6 ^defgh^ ± 0.6
Smuga	0	755 ^hij^ ± 26	90 ^lmn^ ± 2	78.6 ^jk^ ± 2.5	20.6 ^j^ ± 1.3	10.5 ^h^ ± 1.0	38.8 ^fgh^ ± 0.5	10.5 ^bcdefgh^ ± 2.1
	50	780 ^ghi^ ± 47	98 ^hijklm^ ± 7	85.2 ^ghijk^ ± 4.1	26.8 ^defgh^ ± 1.8	12.9 ^cdefgh^ ± 0.9	42.6 ^cdefg^ ± 3.2	12.5 ^abcd^ ± 1.2
	100	653 ^j^ ± 22	82 ^n^ ± 3	66.9 ^l^ ± 1.6	22.9 ^hij^ ± 1.8	10.4 ^h^ ± 0.1	32.3 ^i^ ± 1.2	11.6 ^abcdef^ ± 0.6
PA-15 (Korona)	0	820 ^efghi^ ± 28	87 ^nm^ ± 1	87.2 ^fghij^ ± 2.4	23.8 ^ghij^ ± 1.0	15.0 ^c^ ± 0.9	40.0 ^efg^ ± 0.5	11.4 ^abcdefg^ ± 1.0
	50	885 ^cdefg^ ± 33	104 ^ghijkl^ ± 3	95.1 ^defg^ ± 2.7	26.3 ^efgh^ ± 0.9	13.9 ^cdefg^ ± 0.6	38.3 ^fghi^ ± 0.9	9.3 ^efgh^ ± 0.5
	100	1070 ^b^ ± 5	127 ^cde^ ± 1	108.8 ^c^ ± 0.7	31.3 ^bc^ ± 0.6	14.2 ^cdef^ ± 0.6	46.8 ^bcd^ ± 0.5	13.0 ^abc^ ± 0.6
Plow tillage								
Kora	0	1203 ^a^ ± 43	143 ^b^ ± 5	129.2 ^b^ ± 4.9	35.8 ^b^ ± 1.0	18.7 ^b^ ± 0.6	50.7 ^b^ ± 0.9	13.3 ^ab^ ± 0.5
	50	860 ^cdefghi^ ± 39	116 ^defg^ ± 5	93.4 ^defgh^ ± 4.5	25.8 ^fgh^ ± 1.3	14.9 ^cd^ ± 0.4	41.0 ^defg^ ± 1.2	10.6 ^bcdefgh^ ± 0.5
	100	815 ^efghi^ ± 39	109 ^fghij^ ± 5	85.3 ^ghijk^ ± 3.3	25.3 ^fghi^ ± 1.2	13.6 ^cdefg^ ± 0.6	39.2 ^fgh^ ± 1.7	13.0 ^abc^ ± 1.1
Panda	0	866 ^cdefgh^ ± 57	110 ^fghij^ ± 7	85.2 ^ghijk^ ± 4.2	29.3 ^cdef^ ± 1.4	13.5 ^cdefg^ ± 2.0	41.7 ^defg^ ± 4.2	12.2 ^abcde^ ± 2.2
	50	925 ^cde^ ± 33	131 ^bcd^ ± 6	96.1 ^def^ ± 3.3	29.2 ^cdef^ ± 2.2	15.3 ^c^ ± 1.4	45.3 ^bcde^ ± 2.8	10.9 ^bcdefg^ ± 0.3
	100	964 ^bc^ ± 2	138 ^bc^ ± 1	97.4 ^def^ ± 0.6	32.2 ^bc^ ± 0.2	14.6 ^cde^ ± 0.7	48.3 ^bc^ ± 1.1	13.2 ^abc^ ± 0.0
Smuga	0	817 ^efghi^ ± 45	98 ^hijklm^ ± 5	82.2 ^ijk^ ± 4.2	25.1 ^fghij^ ± 1.9	12.0 ^defgh^ ± 1.2	42.0 ^defg^ ± 2.7	11.3 ^abcdefg^ ± 1.4
	50	746 ^ij^ ± 11	95 ^jklmn^ ± 2	75.4 ^kl^ ± 0.4	21.0 ^ij^ ± 0.5	11.5 ^fgh^ ± 0.2	36.9 ^ghi^ ± 0.9	9.5 ^efg^ ± 0.2
	100	850 ^cdefghi^ ± 23	111 ^fghi^ ± 3	87.5 ^fghij^ ± 2.1	27.8 ^cdefg^ ± 0.6	12.7 ^cdefgh^ ± 0.3	45.5 ^bcde^ ± 0.9	14.2 ^a^ ± 0.8
PA-15 (Korona)	0	806 ^fghi^ ± 11	92 ^klmn^ ± 5	83.4 ^hijk^ ± 2.9	25.0 ^fghij^ ± 1.2	11.0 ^gh^ ± 0.9	40.1 ^efg^ ± 1.8	9.2 ^fgh^ ± 0.6
	50	846 ^defghi^ ± 42	96 ^ijklmn^ ± 6	91.1 ^efghi^ ± 3.8	26.2 ^efgh^ ± 1.2	12.0 ^defgh^ ± 1.4	37.9 ^fghi^ ± 3.2	7.7 ^h^ ± 0.3
	100	897 ^cdef^ ± 15	104 ^ghijkl^ ± 2	97.0 ^def^ ± 1.9	30.5 ^cde^ ± 1.5	13.2 ^cdefgh^ ± 0.6	39.0 ^fgh^ ± 0.8	8.5 ^gh^ ± 0.5

Data are mean values of triplicate determinations ± standard deviation; means with different letters in each column differ significantly (*p* < 0.05).

**Table 3 foods-11-03801-t003:** α-Tocopherol, β-tocopherol, δ-tocopherol, and γ-tocopherol content for four varieties of no-plow tillage and plow tillage common buckwheat, by fertilization doses.

Varieties	Dose of N (kg/ha)	α-Tocopherol (μg/g)	β-Tocopherol Content (μg/g)	δ-Tocopherol (μg/g)	γ-Tocopherol(μg/g)
No-plow tillage					
Kora	0	0.57 ^bcd^ ± 0.01	0.23 ^abc^ ± 0.05	2.53 ^c^ ± 0.34	34.04 ^defg^ ± 0.28
	50	0.57 ^bcde^ ± 0.07	0.31 ^a^ ± 0.08	4.89 ^a^ ± 0.21	41.36 ^b^ ± 0.85
	100	0.37 ^fgh^ ± 0.02	0.17 ^cd^ ± 0.00	2.77 ^c^ ± 0.03	37.33 ^c^ ± 0.50
Panda	0	0.14 ^ij^ ± 0.05	0.18 ^bcd^ ± 0.04	2.24 ^d^ ± 0.16	17.56 ^n^ ± 0.32
	50	0.41 ^efgh^ ± 0.08	0.16 ^cd^ ± 0.00	2.48 ^d^ ± 0.04	31.21 ^hij^ ± 0.83
	100	0.26 ^hi^ ± 0.01	0.16 ^cd^ ± 0.04	1.63 ^e^ ± 0.14	23.86 ^l^ ± 0.50
Smuga	0	Nd ^j^	Nd ^e^	2.08 ^bcde^ ± 1.22	40.35 ^b^ ± 1.17
	50	0.80 ^a^ ± 0.10	0.21 ^abc^ ± 0.02	3.04 ^b^ ± 0.11	40.53 ^b^ ± 1.03
	100	0.75 ^ab^ ± 0.05	0.14 ^cd^ ± 0.02	1.95 ^d^ ± 0.20	34.98 ^de^ ± 0.59
PA-15 (Korona)	0	0.60 ^bcd^ ± 0.05	0.17 ^cd^ ± 0.04	2.32 ^d^ ± 0.15	32.31 ^fghi^ ± 0.89
	50	0.45 ^defg^ ± 0.05	0.14 ^cd^ ± 0.00	2.85 ^c^ ± 0.02	33.19 ^efgh^ ± 0.53
	100	0.68 ^abc^ ± 0.04	0.23 ^abc^ ± 0.05	3.01 ^b^ ± 0.10	47.18 ^a^ ± 0.68
Plow tillage					
Kora	0	0.45 ^defgh^ ± 0.05	0.28 ^ab^ ± 0.02	2.65 ^bcd^ ± 0.40	30.75 ^ij^ ± 0.82
	50	0.62 ^abcd^ ± 0.11	0.18 ^bcd^ ± 0.05	2.25 ^d^ ± 0.19	33.75 ^defg^ ± 0.47
	100	0.56 ^cdef^ ± 0.03	0.18 ^bcd^ ± 0.03	2.25 ^d^ ± 0.34	35.53 ^cd^ ± 0.19
Panda	0	0.68 ^abc^ ± 0.03	0.16 ^cd^ ± 0.03	1.62 ^e^ ± 0.13	32.16 ^fghi^ ± 0.12
	50	0.32 ^gh^ ± 0.07	0.19 ^bcd^ ± 0.01	2.36 ^d^ ± 0.18	32.26 ^fghi^ ± 0.79
	100	0.34 ^gh^ ± 0.09	0.15 ^cd^ ± 0.02	1.70 ^e^ ± 0.05	31.69 ^ghij^ ± 0.18
Smuga	0	0.53 ^cdef^ ± 0.08	0.17 ^bcd^ ± 0.05	2.09 ^d^ ± 0.21	27.51 ^k^ ± 1.12
	50	0.69 ^abc^ ± 0.03	0.15 ^cd^ ± 0.02	2.35 ^d^ ± 0.11	34.72 ^de^ ± 0.36
	100	0.63 ^abcd^ ± 0.06	0.15 ^cd^ ± 0.02	1.52 ^e^ ± 0.06	31.79 ^ghij^ ± 0.58
PA-15 (Korona)	0	0.62 ^abcd^ ± 0.08	0.10 ^de^ ± 0.00	1.68 ^e^ ± 0.01	24.58 ^l^ ± 0.29
	50	0.62 ^abcd^ ± 0.04	0.14 ^cd^ ± 0.01	2.09 ^d^ ± 0.10	29.73 ^j^ ± 0.46
	100	0.27 ^hi^ ± 0.05	0.18 ^bcd^ ± 0.02	1.39 ^e^ ± 0.14	20.22 ^m^ ± 0.75

Data are mean values of triplicate determinations ± standard deviation; means with different letters in each column differ significantly (*p* < 0.05), Nd- not detected.

## Data Availability

The data used to support the findings of this study can be made available by the corresponding author upon request.

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
