# Peer review of "The Lipid-Soluble Bioactive Substances of Fagopyrum esculentum Varieties under Different Tillage and Nitrogen Fertilisation"

_foods, 2022, doi:10.3390/foods11233801_

Round 1

Reviewer 1 Report

The manuscript entitled " The lipid soluble bioactive substances of four varieties of Fag-2 opyrum esculentum according to tillage and fertilisation with 3 nitrogen" aimed to evaluate the effects of fertilizers and nitrogen dosage? on the content of phytosterols, squalene, and tocopherols in four varieties of common buckwheat using GC-MS. The results indicated that the grain of Kora variety contains the highest content of tocopherols. This is meaningful and could provide some reference on the nitrogen fertilization and the cultivation of common buckwheat. This manuscript is structured well and easy to follow, and some minor revisions should be made as follows:

(1)  L213: The results need more scientific interpretation.

(2)  Figure 2(a): it is hard to identify the point in graph.

(3)  L308-309: “no plow tillage cultivation of Kora variety with 50 kg/ha can be used …”, Table 1 Total phytosterol, tocopherol, squalene, and cholesterol content of four varieties of no plow tillage and plow tillage common buckwheat, by fertilization dose also showed the higher content of total tocopherol in grain with no plow tillage cultivation than plow tillage cultivation, why?

Author Response

Rviewer 1:

Thank you very much for your comments. I appreciate your time. Regarding to your comments I did changes:

L213: More information were placed in the text. I characterise more details of the source of nitrogen: ammonium sulphate (50%/50%, NO3-/NH4+) – this information was placed in Methods section:

The first of the sentence was deleted, the second one was changed:“The plow tillage method and doses of nitrogen did not cause any toxicity effect in the plant morphology of four varieties of common buckwheat. The main goal of nitrogen fertilization is to increase the yield of grain [11]. We observed the increase grain yield after nitrogen fertilization. A similarly tendency has been observed by other authors when nitrogen fertilization was increased from 0 to 30 kg·ha-1 N2, giving a final grain yield of about 1.54 t ha-1 [12]., however the authors with higher application of nitrogen observed grain scattering and lodging. Presented research on the grain yield of two cultivars of common buckwheat (Hruszowska and Prego) also showed differences, and the more significant effect of nitrogen fertilizer was observed in the case of Hruszowska variety [12].

Figure 2. The graph resented many points (about 200). The PCA analysis usually it is used for presenting many results in one coordinate system.. The main goal of this analysis is changing from multidimensional system to one dimension. Regarding to the points we can observed Smuga variety without nitrogen fertilizer and Kora variety without fertilizer clearly stand out. Therefore the most important points are Smuga and Kora. Rest of the results is in details placed in other tables.

L308-309: Regarding to your comments, there is no tendency that higher content of total tocopherol was observed in grain with no plow tillage cultivation in comparison to plow tillage cultivation. In the case of Kora variety no differences were observed in Kora with no fertilizer and with 100 kg/ha fertilizer. In the case of Panda variety we did not observe differences in the case of cultivation with 50 kg/ha nitrogen, in Smuga – 100 kg/ha nitrogen no differences. So we did not discuss those relations in this work. Further research in the field of molecular biology should be done to explain those relations.

Reviewer 2 Report

check the pdf file attached

Author Response

Reviewer 2:

Thank you for the attached document with your valuable comments. All suggestions were placed in the main text:

Line 1-3: The title was changed regarding to your right comment: The lipid soluble bioactive substances of four varieties of Fagopyrum esculentum varieties under different according to tillage and nitrogen fertilisation with nitrogen”

Line 18: “tillage” was added, “fertilizers” deleted.

Line 21: and line 24: kg N2 Ha-1 was systemized within all text.

Line 53: The text on tillage in Introduction section was added: “There are several tillage systems: plow tillage, no-till tillage, no tillage. The trend of moving away from plowing, which is energy-consuming and causes soil degradation, is increasingly observed (https://doi.org/10.17221/223/2017-PSE). The effects of tillage different methods on wheat quality shows vary results, however many authors have described its important role regarding to protein content, hectolitre mass, gluten and particle size index of wheat, therefore it seems to be justified to tested other crop cultivation .”

Line 82: To the text “season” was added.

Line 91: Details of tillage were added: “The first factor was soil tillage (plow tillage or lack thereof), conventional tillage applied in autumn: cultivation with a disc cultivator  (2X), plowing with a 4 furrow plow. In spring harrowing with a heavy harrow cultivation with a disc cultivator pre-sowing cultivation with an aggregate. No plow tillage applied in autumn: cultivation with a disc cultivator  (3x). In spring: harrowing with a heavy harrow, cultivation with a disc cultivator, pre-sowing cultivation with an aggregate.”

Line 92: Information about the source of nitrogen was placed in the text: ammonium sulphate (50%/50%, NO3-/NH4+) – this information was placed in Methods section

Line 139, 191, 195: dot was remove throughout all text.

Line 304-308: Some of the information about percent increases for various treatments comparison were added: “The phytosterol content in Kora variety with 50 kg ha-1 of nitrogen was higher by 33% in comparison to crop without nitrogen. In Smuga varieties we observed the increase by 20% of total phytosterols between 100 and 50 kg ha-1 nitrogen fertilizer.”

Reviewer 3 Report

Although the subject might have some merit, the presentation of the results is too confused and confusing. Additionally, there are several errors or misunderstandings. The first is that the Authors describe phytosterols, squalene and tocopherols either as lipids or as soluble substances, but without a clear definition. Tocopherols are liposoluble compounds. Hence, a through presentation of the three groups of substances in the introduction is needed to avoid misunderstandings.

In the introduction, several statements need citations: e.g. lines 40-43; line 43; lines 43-44.

The Foods journal format of the citations is not followed closely, as sometimes the citations are not numbered (e.g. lines 58-59; lines 97-98; lines 261-262; lines 264-265) and do not appear in the References list.

Line 68. Tocols are not lipids. Additionally, the Authors do not address the lipids composition of buckwheat, but only the phytosterols and squalene contents. No fatty acids at all.

Another shortcoming is that in the results, Figure 2, Table 1 and the discussion relevance is given to cholesterol content, but this compound is not mentioned in the aims nor in the Materials and methods. And cholesterol in not a phytosterol nor, as far as I know, is present in buckwheat.

Lines 129.135. Statistical analysis. Please give more details about the factors.

In the Results, please show the results of the ANOVA in a Table. I had problems in understanding the flow of the results because the ANOVA results are not presented. In fact, the Authors state n the summary and also elsewhere that the cultivation method (tilling vs no-tilling) “did not affect the levels of phytosterols, tocopherols and squalene” (Lines 22-23). Hence, they should be considered as a non-factor and not considered in showing the results. Additionally, this would simplify the presentation of Tables 1, 2 and 3, which are too cumbersome. In my opinion, the Authors should carefully check the results of the ANOVA, determine if the two-way interactions are significant and if they are relevant. If not, presenting the results in the Tables should be done considering fertilization as the main trait and merging the results of the four varieties. As stated in the title, tillage and fertilisation are the traits under scrutiny. The Tuckey test should be limited to the three nitrogen levels, and not  spread over 12 results. As the results are actually presented, it is not correct and only leads to confusion.

Lines 214.215. Toxicity effects? Morphology?

Figure 2. (a) the legends of the graph are in Polish; (b) the legends are in Polish; define the abbreviations.

Author Response

Reviewer 3

Dear Reviewer,

Thank you for your valuable comments. During translating the manuscript unfortunately it was changed meaning of sentence, therefore the mistake, which confused readers were improved.

Lines 40-43; line 43; lines 43-44: the citations were filled up.

Lines 58-59; lines 97-98; lines 261-262; lines 264-265: the citation format were improved and lack of citations were added.

Line 68: the information was corrected within all text.

Another shortcoming is that in the results, Figure 2, Table 1 and the discussion relevance is given to cholesterol content, but this compound is not mentioned in the aims nor in the Materials and methods. And cholesterol in not a phytosterol nor, as far as I know, is present in buckwheat.

Next to plant sterols very often are detected small amounts of cholesterol. In tested varieties we detected cholesterol and confirmed the absence by using GC-MS method. I would like to pointed that there is just small amounts, therefore in the aim of the study we did not mentioned about this substances. Usually cholesterol is visible during phytosterols estimation and this information I added to the Method section. (Behrman, E. J., & Gopalan, V. (2005). Cholesterol and Plants. Journal of Chemical Education, 82(12), 1791. doi:10.1021/ed082p1791). 

“Sterols and cholesterol were initially identified by chromatogram comparison of the current samples with previously analyzed samples and other well described in literature oils (sunflower oil, olive oil etc.). Quantitation was carried out using internal standards.” (Figure 1- pick number 7).”

Lines 129.135. Statistical analysis. Please give more details about the factors.

In the Results, please show the results of the ANOVA in a Table. I had problems in understanding the flow of the results because the ANOVA results are not presented. In fact, the Authors state n the summary and also elsewhere that the cultivation method (tilling vs no-tilling) “did not affect the levels of phytosterols, tocopherols and squalene” (Lines 22-23). Hence, they should be considered as a non-factor and not considered in showing the results. Additionally, this would simplify the presentation of Tables 1, 2 and 3, which are too cumbersome. In my opinion, the Authors should carefully check the results of the ANOVA, determine if the two-way interactions are significant and if they are relevant. If not, presenting the results in the Tables should be done considering fertilization as the main trait and merging the results of the four varieties. As stated in the title, tillage and fertilisation are the traits under scrutiny. The Tuckey test should be limited to the three nitrogen levels, and not  spread over 12 results. As the results are actually presented, it is not correct and only leads to confusion.

I would like to apologise for mass in this “Statistical analysis” section. There was conducted one-way analysis of variance (Anova) and if the F was significant the post-hoc (by using Tukey test) was done.  The section was corrected: “All samples were analyzed in triplicate. One-way analysis of variance (Anova) was used. Post-hoc (Tukey’s) test to determine the significance of differences between means was conducted. Dependencies were considered statistically significant at α < 0.05. The median results were presented on a box and whisker plot (P ≤ 0.05). Principal component analysis (PCA) was performed to reduce dimensionality of data and to present the samples in coordinate system. All calculations were carried out with Statistica software (Statistica 13.3, StatSoft, Tulsa, OK, USA).

The F factor was almost in all cases significant, only in the case of cultivation/stigmasterol; cultivation/beta-sitosterol; and cultivation/d7-avenasterol and cultivation/total phytosterol was not significant.

Lines 214-215. The sentence was deleted: “The plow tillage method and doses of nitrogen did not cause any toxicity effect in the morphology of four varieties of common buckwheat. The main goal of nitrogen fertilization is to increase the yield of grain

Figure 2.: The language of legend was changed for English. The abbreviations were explained.
